# Proteomic Landscape of the Mature Roots in a Rubber-Producing Grass *Taraxacum Kok-saghyz*

**DOI:** 10.3390/ijms20102596

**Published:** 2019-05-27

**Authors:** Quanliang Xie, Guohua Ding, Liping Zhu, Li Yu, Boxuan Yuan, Xuan Gao, Dan Wang, Yong Sun, Yang Liu, Hongbin Li, Xuchu Wang

**Affiliations:** 1Key Laboratory of Xinjiang Phytomedicine Resource and Utilization of Ministry of Education, College of Life Sciences, Shihezi University, Shihezi 832003, China; xiequanliang001@163.com (Q.X.); zhuliping0903@163.com (L.Z.); yulixjnu@163.com (L.Y.); yuanboxuan111@163.com (B.Y.); gaoxuan850419@163.com (X.G.); 2Key Laboratory for Ecology of Tropical Islands, Ministry of Education, College of Life Sciences, Hainan Normal University, Haikou 571158, Hainan, China; dingguohuasw@163.com (G.D.); y1yang1y@163.com (Y.L.); 3Institute of Tropical Biosciences and Biotechnology, Chinese Academy of Tropical Agricultural Sciences, Haikou 571101, Hainan, China; wangdanqz2009@126.com (D.W.); sunyong_03119308@126.com (Y.S.)

**Keywords:** natural rubber biosynthesis, mass spectrometry, rubber grass, rubber latex, shotgun proteomics, *Taraxacum kok-saghyz*, two-dimensional gel electrophoresis, visual proteome map

## Abstract

The rubber grass *Taraxacum kok-saghyz* (TKS) contains large amounts of natural rubber (cis-1,4-polyisoprene) in its enlarged roots and it is an alternative crop source of natural rubber. Natural rubber biosynthesis (NRB) and storage in the mature roots of TKS is a cascade process involving many genes, proteins and their cofactors. The TKS genome has just been annotated and many NRB-related genes have been determined. However, there is limited knowledge about the protein regulation mechanism for NRB in TKS roots. We identified 371 protein species from the mature roots of TKS by combining two-dimensional gel electrophoresis (2-DE) and mass spectrometry (MS). Meanwhile, a large-scale shotgun analysis of proteins in TKS roots at the enlargement stage was performed, and 3545 individual proteins were determined. Subsequently, all identified proteins from 2-DE gel and shotgun MS in TKS roots were subject to gene ontology and Kyoto Encyclopedia of Genes and Genomes (KEGG) enrichment analyses and most proteins were involved in carbon metabolic process with catalytic activity in membrane-bounded organelles, followed by proteins with binding ability, transportation and phenylpropanoid biosynthesis activities. Fifty-eight NRB-related proteins, including eight small rubber particle protein (SRPP) and two rubber elongation factor(REF) members, were identified from the TKS roots, and these proteins were involved in both mevalonate acid (MVA) and methylerythritol phosphate (MEP) pathways. To our best knowledge, it is the first high-resolution draft proteome map of the mature TKS roots. Our proteomics of TKS roots revealed both MVA and MEP pathways are important for NRB, and SRPP might be more important than REF for NRB in TKS roots. These findings would not only deepen our understanding of the TKS root proteome, but also provide new evidence on the roles of these NRB-related proteins in the mature TKS roots.

## 1. Introduction

Natural rubber (NR, cis-1,4-polyisoprene) is a biopolymer with high economic value and it is widely used as a strategic raw material in more than 40,000 products [1,2]. More than 2500 plant species can biosynthesize NR [3,4]. However, high-quality natural rubber in viable quantities is only observed in a few plant species, such as the *Para* rubber tree *Hevea brasiliensis* [5,6], Russian dandelion *Taraxacum kok-saghyz* Rodin (TKS, also called the rubber grass) or its close species *Taraxacum brevicorniculatum* [7], *Eucommia ulmoides Oliver* and a guayule shrub *Parthenium argentatum* [8,9]. Currently, the *Para* rubber tree is nearly the only commercial plant to cultivate the exclusive source of NR [5,6]. However, rubber production of the *Para* rubber tree has reached the limit due to the strict climatic requirements for its planting areas [10], little genetic variability, labor cost, and threats of fatal fungal plant diseases [1]. Therefore, it is critical to find an alternative source and a model plant for NR production.

TKS has drawn special attention since the 1940s. It belongs to the Compositae family and originates from the Tekes River Basin near Tianshan Mountain border between Kazakhstan and China [11]. TKS root contains NR ranging from 3% to 28% of total dry weight [1] and its rubber property and molecular characteristics are similar with NR in the *Para* rubber tree [12,13]. TKS grows widely in temperate and cold areas, and its root also contains about 28% of inulin, which is an important material for bioethanol and the food industry [14,15]. With its advantages of high rubber content and quality, wide planting area, a relatively short life cycle, ease of transformation and harvesting method [14,15], TKS is becoming a promising crop for natural rubber production. In addition, TKS is a perennial herb with a relatively simple genome (1.29 Gb), containing 46,731 protein-coding genes [16]. It is easily manipulated for genetic transformation and can be used as an ideal model plant for studying the rubber biosynthesis mechanism [17].

NR is biosynthesized by two pathways: the mevalonate acid (MVA) pathway [18,19] and the methylerythritol phosphate (MEP) pathway [20]. Recently, to determine the protein-based regulation mechanism of NRB, several proteomics have been conducted on the total latex [21,22,23,24,25] and rubber particles [26,27,28,29,30,31] of the rubber tree *H. brasiliensis*. But no proteomic analysis has been performed on TKS, and only one study on determining proteins from the rubber phase and pellet phase of the latex from *T. brevicorniculatum* (TBR) was reported and 278 unique proteins from the one- and two-dimensional gel electrophoresis (1-DE and 2-DE) gels were identified [32]. In order to gain further insight into the protein-based regulation mechanism of rubber biosynthesis, we performed a comprehensive proteomics analysis of the mature roots of TKS at six months (6M) by 2-DE and mass spectrometry (MS), and then identified thousands of proteins by a large-scale shotgun method. These results may deepen our understanding of the root proteome and provide valuable gene candidates for genetic improvement of TKS breeding as a commercial alternative crop.

## 2. Results

### 2.1. Morphological Observation of Rubber Particles in the Main Roots of TKS

Natural rubber in TKS is produced from the latex system of its enlarged roots, and the biomass of the main roots is important for rubber yield. Therefore, the mature 6-month TKS plants with flowers were selected to perform morphological analysis, and the enlarged main roots are highlighted (Figure 1A). The light microscope results revealed that the concentric rings of laticifers as specialized tubular vessels in roots can be detected, and the total number of laticifer vessels is about 137 ± 32 (*n* = 20) in the main roots of 6M TKS. The brown rubber latex granules, known as the specific cytoplasm of laticifer cells, can be clearly detected in the areas of endodermis and cortex in phloem tissues from both the crosscutting and slitting slices of the main roots. All the laticifer cells were considered to originate from the initial cells of vascular cambium to secondary laticifers in paralleling rings (Figure 1B), and most of these laticifer cells seem to connect into tubular vessels in the main roots (Figure 1C). Ultrastructural investigation of rubber particles in TKS root at 6 months was conducted under a transmission electron microscope (TEM). Many spherical or ovoid-ellipsoid rubber particles, ranging from 50 nm to 2000 nm, were examined in the laticifer cells (Figure 1D). We noticed that most rubber particles have a diameter less than 200 nm; they are termed small rubber particles (SRPs). However, rubber particles with diameter larger than 400 nm are fewer in number; they are traditionally called large rubber particles (LRPs). These particles, including SRPs and LRPs, are surrounded by a monolayer membrane (Figure 1D), in which many enzymes related to NRB are anchored or combined with each other [33].

### 2.2. Establishment of a Visual Two-Dimensional Gel Electrophoresis (2-DE) Proteome Map and Mass Spectrometry (MS) Identification of High Abundance Proteins in Taraxacum Kok-saghyz (TKS) Roots

To further obtain the protein accumulation profiling, the total proteins from the 6M TKS roots were extracted and performed 2-DE to separate the total proteins. Finally, a high-resolution 2-DE gel was obtained and a visual proteome map was established (Figure 2). In this reference 2-DE gel, 428 ± 15 protein spots, ranged from pH 4 to pH 7, were detected from three biological replicates (Appendix A). We manually excised all abundant protein spots from the 2-DE gel, and finally identified 371 protein species by MALDI TOF/TOF MS. These proteins occupied 84.38% volume of all the detected spots in the 2-DE gel and belonged to 231 gene products or named unique proteins (Appendix A).

We compared the theoretical and experimental ratios of molecular weight (M*r*) and isoelectric point (*p*I) of the 371 proteins identified from the 2-DE gel, and presented their ratios as radar axis labels (M*r* for radial value; *p*I ratio for annular value) in radial chart (Figure 3A). Most proteins showed similar radial and annular values, and their ratios for theoretical and experimental M*r* and *p*I are near line 1.0. However, some proteins demonstrated differentially experimental M*r* and *p*I values in the 2-DE gel (Figure 2). In the proteomics study based on 2-DE gel, they are called different protein isoforms or protein species. In this study, 90 unique proteins were identified from at least two different spots, and these proteins contained 229 spots in the 2-DE gel. Among them, 56 proteins were identified from 112 spots, and they contained two protein isoforms in the 2-DE gel. Twenty-two proteins had three isoforms, and nine proteins were determined from 36 spots with four isoforms (Appendix A). There are three unique proteins containing five isoforms; they are aldolase (spots 305, 313, 314, 403 and 405), enolase (spots 94, 100, 106, 150 and 158) and cytosolic glyceraldehyde-3-phosphate dehydrogenase (GAPC, spots 12, 163, 176, 272 and 275).

It is noteworthy that these protein isoforms show the same theoretical M*r* and *p*I values. Most of their M*r* values are ranged from 20 kDa to 60 kDa, and their *p*I points are mainly distributed from pH 5.0 to 7.0 (Figure 3B). However, their experimental M*r* and *p*I values are different to each other, and most protein isoforms showed a different *p*I value in the 2-DE gel (Figure 1). These results indicated that these proteins could have post-modifications that resulted in protein forms/species in the 2-DE gel-based proteomics.

We further determined the protein abundance in the 2-DE gel and the most abundant 20 protein species are highlighted (Figure 3C). The results demonstrated that, based on the spot volume, the top-20 protein spots occupied almost a half of abundance volume (43.84%) for all proteins in the 2–DE gel of 6M TKS roots. The most abundant protein spots were identified as the protein species or isoforms of catalase (spot 355, 6.57%; spot 2, 3.38%), ferritin (spot 357, 3.77%; spot 359, 2.90%), actin (spots 142 and 354), actin (spots 142 and 354), GAPC2 (spots 4 and 118), and the subunit of ATPase (spots 11 and 344). Among them, the most abundant spot was identified as catalase-like isoform X1 (spot 355), followed by allene oxide cyclase (spot 346) and eukaryotic translation initiation factor 5A (eIF-5A, spot 361) (Appendix A).

### 2.3. High-Throughput Shotgun Proteomic Analysis of the 6M Roots of TKS

As proteins with very low abundance or out of the pH range of immobilized pH gradient (IPG) strips are difficult to separate and identify by 2-DE-based proteomic method, a shotgun analysis of the proteins was further performed. The proteins were extracted from the 6M TKS roots, and three biological repeats (R1, R2 and R3) were conducted to obtain a comprehensive proteomic profile. A total of 5205, 5323 and 5654 unique proteins were successfully identified from R1, R2 and R3, respectively (Appendix A). Among them, 7481 proteins were identified from at least one independent shotgun experiment, and 5156 proteins were identified from at least two experiments. There were 3545 shared proteins in the three experiments (Figure 4A), and only these shared proteins from three experiments were considered as the 6M TKS root proteins produced by the shotgun proteomic method in the following study. We found 184 shared proteins in the 2-DE and shotgun proteomic methods. There were 3361 and 47 specific proteins in the shotgun and 2-DE methods, respectively. These results indicated that, although the shotgun method can produce large amounts of proteins, the traditional 2-DE gel-based proteomics also generates a few specific proteins from the 6M TKS roots (Figure 4B; Appendix A). Our special interest is focusing on the potential biological functions of these shared proteins in TKS roots.

### 2.4. Pathway Analysis of the Identified Proteins in 6M TKS Roots

To gain insight into the functional categories of the 3592 proteins (including the 231 and 3545 proteins identified by 2-DE and shotgun, respectively), we performed gene ontology (GO) classification and the enriched outputs of their biological process, cellular component and molecular function are presented (Figure 5). The GO enrichment revealed that many proteins were localized in membrane-bounded organelle, and they were mainly involved in metabolic process with catalytic activity or binding ability (Appendix A). At the biological process level, 13 main processes were detected. Among them, the largest amount containing 1520 proteins were involved in the metabolic process, followed by 1033 proteins involved in the cellular process and 695 proteins in the single-organism process. A total of 302 proteins were considered to respond to external stimulus. At subcellular level, nine components were observed. Among them, almost half including 866 proteins were localized to the cell part. Many proteins were localized into organelle membrane and macromolecular complex. In molecular function classification, seven pathways were determined. Among them, the most portion including 1688 proteins were taken part into catalytic activity, followed by 1325 proteins with binding ability. There are 107 proteins with transporter activity and 101 proteins with structural molecule activity (Figure 5; Appendix A). These different GO term distributions in TKS roots are related to their biological functions and indicate some important biological processes for secondary metabolite in the 6M TKS roots.

To further investigate the biological functions of the identified 3592 proteins, KEGG pathway analysis was performed using the BLAST2GO program. These proteins were clustered into 19 main pathways (Figure 6A), including translation process (616 proteins), carbohydrate metabolism (583 proteins), folding, sorting and degradation (440 proteins), global and overview metabolism (409 proteins), amino acid metabolism (380 proteins), transport and catabolism (342 proteins), lipid metabolism (308 proteins), etc. These results demonstrated that the main proteins in TKS roots were involved in translation and carbohydrate metabolism. Furthermore, 129 sub-pathways were enriched in KEGG annotation (Appendix A), and the top 20 sub-pathways are highlighted (Figure 6B). Among them, the most abundant pathway containing 240 proteins is carbon metabolism. The second contains 224 proteins with ribosome activity. The third sub-pathway has 213 proteins for biosynthesis of amino acids, followed by 183 proteins for endocytosis, 182 proteins in the spliceosome, and 172 proteins in the endoplasmic reticulum. Proteins involved in starch and sucrose metabolism (164 proteins), glycolysis (121 proteins), phenylpropanoid biosynthesis (119 proteins), and amino sugar and nucleotide sugar metabolism (116 proteins), are also enriched in these KEGG sub-pathways (Appendix A). These proteins are crucial for biosynthesis of isoprenoids in many plants, especially in natural rubber-producing plants [16].

### 2.5. The Proteins Involved in Natural Rubber Biosynthesis in TKS Roots

We checked the identified 3592 proteins from the 6M TKS roots and paid special attention to the proteins and enzymes involved in NRB. Our proteomic data demonstrated that a total of 58 unique proteins or gene products were identified at least one time by the three shotgun experiments (Appendix A), and four of them, named ACAT7 (spots 157, 183 and 424), HMGR1 (spot 146), DXS3 (spot 257) and DXS8 (spots 199 and 204), were also identified from the 2-DE gel of 6M TKS roots (Figure 1; Appendix A). Twenty kinds of proteins were determined from the 58 unique gene products (Appendix A), and these proteins are involved in both MVA and MEP pathways for NBR in TKS roots (Figure 7). Genomic sequencing data demonstrated that both cytosolic MVA and plastidic MEP pathways are present for NBR, and a total of 102 NBR-related genes have been determined in the genome of TKS [16]. Our proteomic data demonstrated 22 unique proteins were involved in the MVA pathway and 13 proteins in MEP pathway in TKS roots (Figure 7; Appendix A). In the MVA pathway, the identified 22 unique proteins belonged to six kinds of essential enzymes for rubber biosynthesis; they are ACAT, HMGS, HMGR, MVK, PMVK and MVD. Among them, seven members for ACAT (from ACAT2 to ACAT8), six members for mevalonate kinase (MVK 1, 2 and 7–11), and five members for phosphomevalonate kinase (PMVK 1 and 2–6), were positively identified from TKS roots by the shotgun method (Appendix A). Our proteomics results also showed that, by contrast with rubber tree latex, TKS contains the MEP pathway for NRB in its roots. In the MEP pathway, seven kinds of proteins were identified from 13 unique proteins, including six family members for deoxyxylulose-5-phosphate synthase (DXS 1–3 and 8–10) and two members for D-xylulose 5-phosphate reductoisomerase (DXR 1 and 2).

In the following step for initiation of the synthesis of isopentenyl pyrophosphate (IPP) to form rubber molecule polymer, which is termed initiator synthesis, four crucial enzymes, including isopentenyl-diphosphate delta-isomerase (IPI 2–4), geranylgeranyl pyrophosphate synthase (GGPS 1–3), geranylgeranyl diphosphate synthase (GPS 3, 5 and 7) and farnesyl diphosphate synthase (FPS 1), were identified from 10 unique proteins.

In the final rubber elongation process, cis-isoprene transferase (CPT) can help GGPP to generate natural rubber hydrocarbons contain different length of carbons. Four kinds of proteins, named CPT, SRPP, REF, and HRBP, which is a Nogo-B receptor as a HRT1-REF bridging protein, are considered to play crucial roles for natural rubber elongation. In this proteomic study, we identified the above four members from 12 unique proteins. Among them, eight SRPP members (SRPP 1–7 and 9) and two REF members were determined from the 6M TKS roots.

We checked the annotated genomes of the rubber tree *H. brasiliensis* and the rubber grass *T. kok-saghyz*, and found 85 and 106 NRB-related proteins from *H. brasiliensis* and TKS genomes, respectively. It is noteworthy that 58 out of 106 NRB-related unique proteins were identified from the 6M TKS roots, and 13 out of the identified 58 unique proteins were crucial members in the MEP pathway (Appendix A). These results revealed that both MVA and MEP pathways are important in controlling NRB and rubber production in the mature TKS roots.

## 3. Discussion

### 3.1. The First Visual Proteome Based on 2-DE Gel Demonstrated Many Protein Isoforms in the Mature TKS Roots

The ease and availability of 2-DE combined MS is a classical standard for proteomics study, and it is still a valuable tool to provide a visual proteome for plants [34]. Large amounts of proteomics studies have been performed in many plant species, however, only several proteomic research works were reported on different latex components of the *Para* rubber tree [2,35], and only one proteomic analysis was performed to identify 278 unique proteins from the rubber particle phase and pellet phase of the *T. brevicorniculatum* latex [32]. Our 2-DE gel-based proteomics of TKS roots resulted in 371 abundant protein species (Figure 2; Appendix A), but the protein profiles are very different to that in *T. brevicorniculatum* latex. Compared with the 2-DE gels for a clear aqueous phase and a yellowish pellet phase of the fractionated samples from *T. brevicorniculatum* latex [32], our 2-DE gel for the mature TKS roots is much clearer and contains more protein spots (Figure 2). Among the identified proteins, 90 unique proteins contained different protein isoforms, which have different experimental M*r* and *p*I values on the 2-DE gel (Appendix A), probably due to the existence of modification protein variants.

In the past decade, only 1208 proteins were identified from all the rubber-producing plants, and only several proteomics studies were performed on the rubber particles [36]. In a previous high throughput proteomics study, a total of 186 proteins were positively identified from rubber particles by shotgun tandem MS [28]. After removal of the protein bands corresponding to the RP-abundant REF and SRPP, 137 protein species including 115 unique proteins were identified in a LC-MS/MS based proteome analysis [31]. By investigating the protein profile between LRPs and SRPs, 53 protein spots corresponding to 22 gene products were detected to have significant difference. Among them, most up-regulated proteins in SRPs were identified as SRPP, HMGS, phospholipase D, ethylene response factor, eukaryotic translation initiation factor, etc., but the most abundant proteins in LRPs were REF, glucanase and several hypothetical proteins [27]. Interaction network analysis of rubber particle proteomics revealed the formation of the protein complex consisting of HRT1, REF and HRBP might play crucial roles as a NRB machinery [31], and these proteins are associated with the endoplasmic reticulum [37]. 

Recently, more than 1600 proteins were identified from total latex of the *Para* rubber tree by using isobaric tags for relative and absolute quantitation (iTRAQ) method [2] and 1839 unique proteins were determined by LC-MS/MS from the whole translated draft genome of the rubber tree *H. brasiliensis* [24]. Our high throughput shotgun proteomics produced 7481 proteins from at least one independent shotgun experiment and 3545 shared proteins in the three experiments from the 6M TKS roots (Figure 4B; Appendix A), which covered almost all the previously identified proteins from the latex-producing plants. To our best knowledge, it is the first visual proteome based on the combination of 2-DE gel and high throughput proteomics methods, and the results may help us to deepen our understanding of the roles of protein isoforms in the mature TKS roots.

### 3.2. Large-Scale Shotgun Proteomics Landscape Revealed both Mevalonate Acid (MVA) and Methylerythritol Phosphate (MEP) Pathways are Important for Natural Rubber Biosynthesis (NRB) in the Mature TKS Roots

Natural rubber is an isoprenoid polymer that is synthesized on the rubber particles in many latex-producing plants [16]. Although more than 2500 plants can biosynthesize natural rubber [31], the *Para* rubber tree *H. brasiliensis* is the only plant commercially cultivated to produce natural rubber for industry [2]. Rubber latex is a kind of specialized cytoplasm of laticifer cells in the bark phloem of the rubber tree, and natural rubber, as an elastomer with physical and chemical properties, cannot be fully matched by synthetic rubber [31]. Annotation of the rubber tree genome revealed that although 22 MEP genes can be identified, only two DXS genes (*DXS7* and *DXS10*) show substantial and preferential expression in the rubber latex; on the other hand, at least one gene for the identified 18 enzymes in MVA pathway shows latex-biased abundant expression [10]. Proteomics analysis also supported that MVA pathway is more important than MEP pathway for NRB in the latex of the *Para* rubber tree [2,24,25,29]. 

Similar results were observed in the recently assembled TKS genome, which contains 102 rubber biosynthesis-related genes. Among them, 40 genes in the MVA pathway and 23 genes in the MEP pathway were identified, respectively. Comparison of gene expression level proved that, for each process in the MVA pathway in TKS, at least one enzyme shows a predominant gene expression level in latex and roots, but most genes in the MEP pathway have a medium or low expression level in the TKS latex [16]. These results indicated that it is the MVA pathway rather than MEP pathway that is involved in the NBR in the rubber latex of both the rubber tree *H. brasiliensis* and the rubber grass TKS. 

However, our comprehensive proteomics data showed that many proteins involved in both MVA (22 unique proteins) and MEP (unique 13 proteins) pathways have been positively identified from the mature TKS roots (Figure 7; Appendix A). Among the identified 13 MEP-pathway proteins, two unique proteins named DXS 3 (spot 257) and DXS 8 (spots 199 and 204) are visible on the 2-DE gel (Figure 1), and their relative abundance is very high (Appendix A). These proteomic data suggest that, by contrast with that in the inner bark of the *Para* rubber tree [10], both the MEP pathway and the MVA pathway might be the main source of IPP, and MVA pathway is also crucial for rubber biosynthesis in the mature TKS roots.

### 3.3. Almost All NRB-Related Proteins Can Be Identified From the Mature TKS Roots

In the *Para* rubber tree, NRB is known to begin with IPP synthesis in MVA pathway [38,39], which is a cytosolic pathway for rubber biosynthesis [40]. In the early steps of the MVA pathway, ACAT is important for generating acetoacytyl-CoA, then HMGS and HMGR activate the supply of mevalonate substrates [2,41]. ACAT can catalyze a Claisen-type condensation of two acetyl-CoA units to form acetoacetyl-CoA, which is the first step in the MVA pathway and is also important for providing the malonyl-CoA substrate for the biosynthesis of fatty acids [42]. This enzyme was found to mainly accumulate in the mature TKS roots (Appendix A), but it could not be detected in total latex of the *Hevea* rubber tree [2]. Through a series of enzymatic reactions, acetyl-CoA can be catalyzed by ACAT to form IPP [43]. Four ACAT genes were determined in the *Hevea* rubber tree [10] and eight ACAT genes were found in the TKS genome [16]. Overexpression of *AtACAT* results in the increased accumulation of ACAT in the latex of *T. brevicorniculatum*, and ultimately increased the pentacyclic triterpene and sterol levels [44]. Our proteomic results demonstrated that seven (ACAT 2-8) out of the eight ACAT members can be identified by the shotgun proteomic method, and one member (ACAT 7, spots 157, 183 and 424) can also be detected on the 2-DE gel (Figure 2), indicating that ACAT 7 is high-abundant in TKS roots. 

In the following steps, HMGS and HMGR activate the supply of mevalonate substrates [2,41]. HMGS, the second enzyme in the MVA pathway, catalyzes aldol-type condensation of acetoacetyl-CoA and acetyl-CoA to produce HMG-CoA, and it is located as a cytosolic protein [45,46]. In our proteomic data, one HMGS member (HMGS 1, evm.model.utg 270.25; spot 146) was identified from both high-through and 2-DE gel-based proteomics methods (Figure 7; Appendix A). Furthermore, HMG-CoA can be converted to MVA by HMGR [47]. HMGR controls the carbon flow and metabolic reaction rate in MVA [42], and it is also a critical player for the regulation of triterpenoid metabolism [38,48]. It was reported that ethylene cannot activate the enzyme activity of HMGR [49]. Five gene members have been characterized in the genome of the *Para* rubber tree [10], but only HMGR1 gene was found to be induced by ethylene stimulation [49]. In the TKS genome, 12 HMGR members were determined, and only two out of them (*TkHMGR*1 and *TkHMGR*2) were predominantly expressed in roots, with the highest expression level in latex [16]. In this proteomic study, two members of HMGR (HMGR1 and HMGR12) were detected from TKS roots by the shotgun method (Appendix A). Based on these genomic and proteomic results, we consider that HMGR1 might be the most important player for NRB in TKS roots.

In the following steps, mevalonate kinase (MVK) converts mevalonate into isopentyl pyrophosphate and phosphomevalonate kinase (PMVK) catalyzes mevalonate-5-phosphoate to form mevalonate-5-pyrophosphate [50]. In *Hevea*, three MVK genes were determined [10]; and in TKS genome, 11 MVK genes were observed [16]. In our proteomic data, we identified eight MVK protein members from the predicted 11 MVK genes, and five PMVK members from the six PMVK genes (Appendix A). Finally, MVD catalyzes mevalonate-5-pyrophosphate to form IPP [49]. Two MVD members were determined from both *Hevea* and TKS genomes [16], and in this proteomic study, they were identified from the mature TKS roots (Appendix A). These results revealed that enzymes involved in the conversion of mevalonate into isopentyl pyrophosphate and diphosphate to form IPP are important for NRB in the mature TKS roots.

The MEP pathway is mainly occurred in the plastid and it is an alternative IPP synthesis pathway [51,52]. Recently, the plastidic MEP pathway has also been considered as a possible route for NRB [53,54,55]. The ^13^C-labelling of *Hevea* seedlings revealed that MEP pathway mainly contributes IPP for carotenoid biosynthesis [38]. Genomic analysis demonstrated that all genes involved in the MEP pathway can be detected in both *Hevea* and TKS, but most of the MEP-related genes have a medium or low expression level in the latex of the *Hevea* rubber tree [10] and TKS [16]. However, our proteomics results demonstrated that 13 unique proteins involved in the MEP pathway were determined in the TKS roots; these proteins are DXS, DXR, CMS, CMK, HDS and HDR (Appendix A). These results indicated that the MEP pathway may also contribute IPP for NRB in the mature TKS roots.

In the initiator synthesis process, GPS and IPI catalyze IPP and DMAPP to form GPP, and FPS to produce farnesyl diphosphate by adding IPP onto GPP to form polyisoprene [40,56], then GGPS and GPS catalyze farnesyl diphosphate to form geranylgeranyl diphosphate [31,42]. Our proteomics data showed that four crucial enzymes, named IPI, GGPS, GPS and FPS, can be identified from the TKS roots.

Then, in the natural rubber elongation process, cis-isoprene transferase (CPT), also named *Hevea* rubber transferase, catalyzes multiple isoprene units (C5H8) and polymerizes into IPP long chain molecules and determine the size of the rubber molecule [46,57]. It is closely associated with the rubber particle membrane and is widely known to be a key enzyme in rubber biosynthesis [28,31]. Eleven CPT members were determined in the *Para* rubber tree genome [10], but only CPT2 shows rubber transferase activity [40], and a rubber transferase activator (RTA) or CPT-like protein (CPTL) can interact with CPTs to activate NRB in the *Hevea* rubber latex [57]. In the TKS genome, eight CPT members and two CPT-like proteins (CPTL1 and CPTL2) have been characterized, and *TkCPT1, TkCPT2* and *TkCPTL1* genes play a critical role for the elongation of rubber polymers in the latex and roots of TKS [16]. However, our proteomics data only identified CPT1 in the TKS roots (Appendix A), indicated CPT1 may be the crucial player for NRB in TKS roots. 

In the final elongation process, three rubber particle membrane binding proteins, named REF, SRPP and a Nogo-B receptor or HRT1-REF bridging protein (HRBP), are widely known to play crucial roles [35], and they were positively identified from the TKS roots by our proteomics study (Figure 7). Among them, REF is anchored inside rubber particle membrane by its auto-assembly ability [42], whereas SRPP largely covers rubber particle surface in an oriented anisotropic manner [58]. REF is also suggested to be a conservative regulator of lipid droplets and it is related to form an intracellular structure [59,60]. However, proteomics of the detergent-washed rubber particles identified none lipid droplet proteins [28,31], indicated that the maturation of rubber particles might be different from that in lipid droplets. Histochemical localization indicated that rubber biosynthesis capability in *Hevea* laticifer is mostly concentrated in SRPs, and SRPP, a rubber biosynthesis-related protein that accumulated mainly in SRPs, is more important than REF for rubber biosynthesis in the *Hevea* rubber tree [48]. Five SRPP genes (*TbSRPP 1-5*) were found in TBR [7]. Among them, except for *TbSRPP 2* [61], four members can enhance NRB. But the average molecular weight of rubber hydrocarbons has not been significantly affected by these genes [62]. Overexpression of *TbSRPP3* only resulted in a slight change in the content of rubber in TBR root [63]. In this proteomics of TKS roots, eight out of 10 SRPP members were identified (Appendix A). Different with that in *Hevea* genome, which contains eight REF members, only two REF members have been characterized in the TKS genome [16]. Our proteomics data also identified the two REF members in the mature TKS roots (Appendix A). These published results, as well as our proteomics of the 6M TKS roots, revealed that SRPP might be more important than REF for rubber biosynthesis in the mature roots of TKS, and more attention should be paid on determination of the detail roles of SRPP in TKS roots in future.

## 4. Materials and Methods 

### 4.1. Plant Growth Conditions and Root Collection

TKS seeds were collected from the Tekes River basin in Xinjiang, China. Following germination, seedlings were transplanted into the pots in a greenhouse of Shihezi University (Shihezi, China), containing nutritive soil and vermiculite with a ratio of 5/3, and a half strength of Hoagland liquid medium was irrigated with 55% relative humidity at 22 °C in the light and 18 °C in the dark. After growing for 6 months (termed as 6M roots), the main roots from plants were collected, rinsed with tapped water, blotted dry on a filter paper, and the middle parts were dissected into approximately 0.5 cm-thick slices, and then frozen in liquid nitrogen for further study. 

### 4.2. Morphological Observation of Rubber Particles in TKS Roots

For histochemical staining, a 0.5 cm long section from the middle parts of the fresh main roots of TKS were fixed in 4% glutaraldehyde (0.1 M phosphate buffer, pH 7.5) at room temperature, then treated with bromine and iodine in glacial acetic acid as described [64]. Part of the sample was dehydrated through a graded series of ethanol, and embedded in paraffin. Sections (12 μm thickness and area of 5 mm × 5 mm) were cut with microtome (Leica Microsystems, Bannockburn IL, Germany) stained with mercury–bromophenol blue, which is effective in showing the latex protein [65]. Then, the slides were examined under a leica DMLN electron microscope (leica, Wetzlar, Germany). TKS laticifer cells in the root sections could be recognized due to iodine-bromine treatment of the rubber in the laticifers became deep brown or dark color. Another part of the sample was dehydrated in a range of acetone (30%, 50%, 70%, 90%, 1 h for once and pure acetone 2 h for twice). Then it was permeated with acetone and Epon 812 embedding agent and pure Epon 812 embedding agent permeates overnight as described [33]. After that, the samples were polymerized in the oven, flattened and cut into semi-thin slices. Observed under an optical microscope, the laticifers could be recognized by tracing rubber inclusions, which is brown with iodine-bromine staining [65]. For electron microscopy, samples were cut into a smaller size, immediately fixed in the glutaraldehyde solution at 4 °C for 24 h, and then fixed in phosphate buffer (pH 7.2, 2% OsO4 in 0.1 M) for 6 h at room temperature. Ultra-thin sections, double stained with uranium acetate and lead citrate again. A transmission electron microscope (TEM, Hitachi JEM-1230, JEOL, Japan)), country was used to examine these stained slices.

### 4.3. Protein Extraction and Two-Dimensional Gel Electrophoresis

Total proteins were extracted from TKS roots using a modified BPP phenol method as described [22]. The washed protein pellets were air-dried for 10 min and dissolved with the Lysis buffer (30 mM Tris-HCl, pH 8.5, 2 M thiourea, 7 M 17 urea, 4% 3-[(3-Cholamidopropyl) dimethylammonio] propanesulfonate (CHAPS)). Protein concentration was determined by the Bradford assaying, and BSA was used as the protein standard. For 2-DE assay, 1300 μg of proteins were diluted to 450 μL with lysis buffer and loaded onto a 24 cm, linear pH gradient 4–7 IPG strips (GE Healthcare, Uppsala, Sweden). Then, they were rehydrated for 24 h at 22 °C. Isoelectric focusing system and gel electrophoresis were performed as described [26]. The gels were stained with Coomassie Brilliant Blue G-250, and analyzed by the ImageMaster 2-D Platinum Software as described [2]. Biological variation analysis module was employed to identify the accumulated protein spots in roots samples with statistically significant (confidence above 95%, *p* < 0.05). Three biological replicates for each separation were conducted and the three typical 2-DE gels are presented in the Appendix A. 

### 4.4. Identification of Protein Spots via MALDI-TOF/TOF-MS

The main protein spots on the 2-DE gel of 6M roots were manually excised from 2-DE gels and subjected to perform an in-gel digestion with the modified bovine trypsin (Cat. 11418025001, Roche, Basel, Switzerland) as described [64]. After digestion, mass spectra of trypsin-digested peptides map fingerprinting were obtained on an AB 5800 MALDI-TOF/TOF mass spectrometry (AB SCIEX, Foster City, MA, USA) with a laser wavelength of 349 nm as described [2]. Ten main spectra were selected for MS/MS analysis and searched for a self-constructed database derived from the original TKS genome. The nucleotide and protein sequences were deposited in the Genome Warehouse (GWH; http://bigd.big.ac.cn/gwh/) under the accession number PRJCA000437 (https://academic.oup.com/nsr), which includes 46,731 gene sequences [16]. Protein matches were considered as a positive identification with a Mascot score higher than 75 and at least two matched peptides, meanwhile individual ion score > 31 (*p* < 0.05). Detailed information for the identified proteins is provided as the Appendix A). 

### 4.5. Large-Scale Shotgun Proteomics of TKS Roots

Approximately 300 μg proteins from the main roots of TKS were reduced and alkylated by dithiothreitol and iodoacetamide, then digested by trypsin. Enzymic hydrolysates were separated using a Pepmap C18 column (Thermo Fisher, USA) following a 65 min 5%–35% organic gradient by an UltiMate 3000 instrument (Thermo Scientific, Rockford, USA). Fifty washed fractions were collected in 1.5 mL/min tubes. Fractions were dehydrated and merged into 15 fractions, and these samples were subjected to a high-resolution mass spectrometer Triple TOF 6600 system (AB SCIEX). Peptides were automatically selected by ProGroupTM algorithm and ProteinPilot™ software V5.0 (AB SCIEX) to calculate the error factor (EF), reporter peak area and p value as described [30]. A protein database was established by the target protein sequences (https://academic.oup.com/nsr) for the corresponding species. Protein with an unused score > 1.3 (confidence ≥ 95%) was considered as a positive identification. Finally, an in-house BlastP search of the UniProt database was performed for each protein to identify its homologues and potential functions.

### 4.6. Functional Classification, Hierarchical Clustering and Pathway Analysis of All the Identified Proteins From TKS Roots

The sequences of all the identified proteins were search against the UniProt database (http://www.uniprot.org/) to confirm their functional annotations. The identified proteins were then classified by Gene Ontology (GO, http://www.geneontology.org/) as described [66]. The GO terms were further determined by the WEGO software (http://wego.genomics.org.cn) on biological process, molecular functions and cellular components. Finally, these proteins were performed KEGG pathway analysis (http://www.genome.jp/kegg) to determine their molecular reaction networks.

## 5. Conclusions

In this study, we obtained the first visualization proteome profiles of mature TKS roots. Combination of 2-DE and shotgun proteomics leads to more than 3,000 unique proteins in TKS roots. Functional category of all TKS proteins revealed that most proteins were involved in carbon metabolic process with catalytic activity in membrane-bounded organelle, followed by proteins with binding ability, transportation activity, starch and sucrose metabolism, and phenylpropanoid biosynthesis. They are important for biosynthesis of isoprenoids in natural rubber-producing plants. Fifty-eight NRB-related proteins were identified, and they are involved in both MVA and MEP pathways. Our results indicated the MEP pathway is also important by contributing IPP for rubber biosynthesis in TKS roots. Almost all NRB-related proteins can be identified, and several members of many key proteins, including ACAT, HMGR, MVK, PMVK, DXS, DXR, GPS, FPS, IPI, REF and SRPP, were determined from the mature TKS roots. Future work should pay more attention to determination of the detailed roles of different members of these NRB-related proteins in the mature TKS roots.

## Figures and Tables

**Figure 1 ijms-20-02596-f001:**
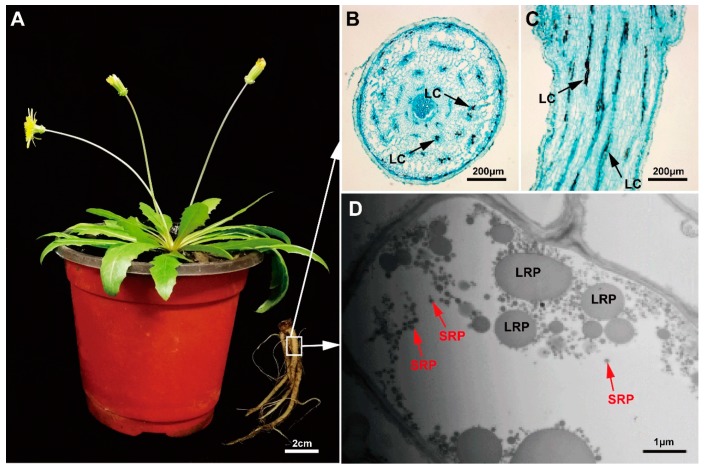
Morphological observation of the laticifer cells and rubber particles in the main roots of *Taraxacum kok-saghyz* (TKS). A six months old TKS plant and its main roots are highlighted (**A**). The laticifer cells containing brown rubber can be examined under the crosscutting (**B**) and slitting (**C**) slices of the 6M TKS roots. Black arrows indicate the positions of typical laticifer cells. Both small rubber particles (SRP) and large rubber particles (LRP) in the main roots of 6M TKS can be observed under a transmission electron microscope (TEM) (**D**).

**Figure 2 ijms-20-02596-f002:**
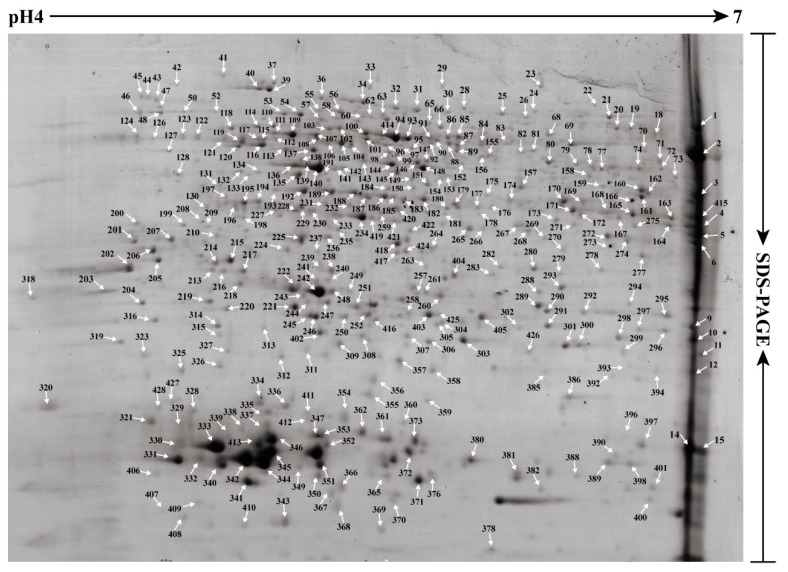
The reference proteome map and MS identification of abundant proteins in 2-DE gels of the 6M TKS roots. The proteins extracted from the 6M mature TKS roots were examined with two-dimensional gel electrophoresis (2-DE) to produce a high-resolution reference proteome map. The 371 major protein spots marked with numbers were positively identified by MALDI TOF/TOF MS and their positions are indicated by white arrows. The detailed identities of these proteins are listed in Appendix A.

**Figure 3 ijms-20-02596-f003:**
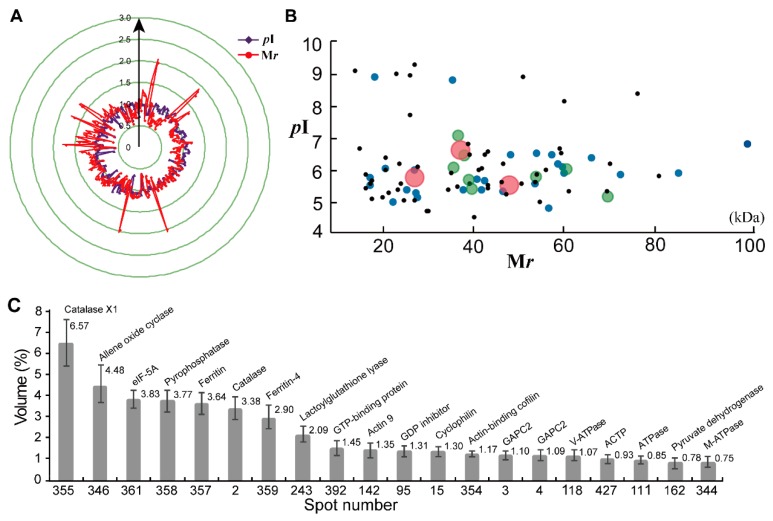
Distribution of the identified proteins on the reference 2-DE gel of 6M TKS roots. The theoretical and experimental ratios of M*r* and *p*I for the identified 371 are presented in radial charts with radial and annular radar axes (**A**). The theoretical distribution of M*r* and *p*I for the protein isoforms is highlighted (**B**). These protein isoforms were independently identified from two (black circle), three (blue circle), four (green circle), and five (red circle) spots on the reference 2-DE gels of the 6M TKS roots, respectively. The most abundant 20 proteins in the 2-DE gel of 6M roots are highlighted (**C**). The detailed identities for these protein isoforms are listed in Appendix A.

**Figure 4 ijms-20-02596-f004:**
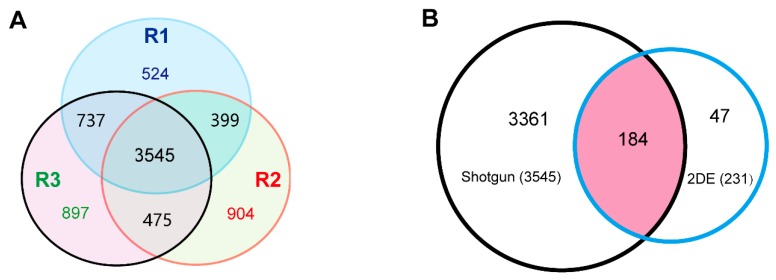
High-throughput proteomics analysis of the 6M TKS roots. Three shotgun proteomics were performed to determine the large-scale protein profiles in the 6M roots, and finally identify 3545 shared proteins in the three experiments (**A**) The blue, yellow, and pink circles represent for the first, second, and third biological replicates. Among these proteins, 184 proteins could also be detected from the 2-DE gel (**B**). The pink area represents for the 184 shared proteins that were identified by both the 2-DE and shotgun based proteomic methods.

**Figure 5 ijms-20-02596-f005:**
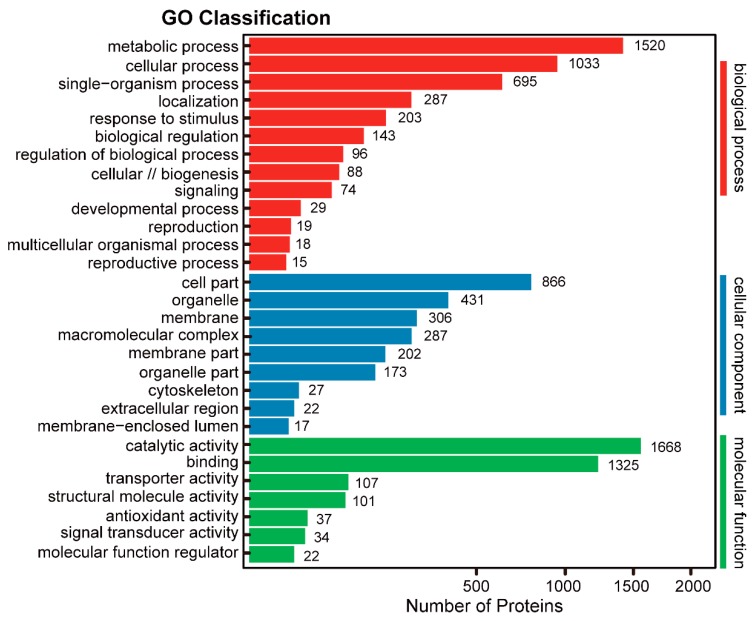
GO classification of the identified proteins. The 3592 proteins identified from the 6M roots by both shotgun and 2-DE methods were performed GO analysis and these proteins were classified into 3 main categories including biological process, cellular component and molecular function. The number of proteins denotes those with GO annotations.

**Figure 6 ijms-20-02596-f006:**
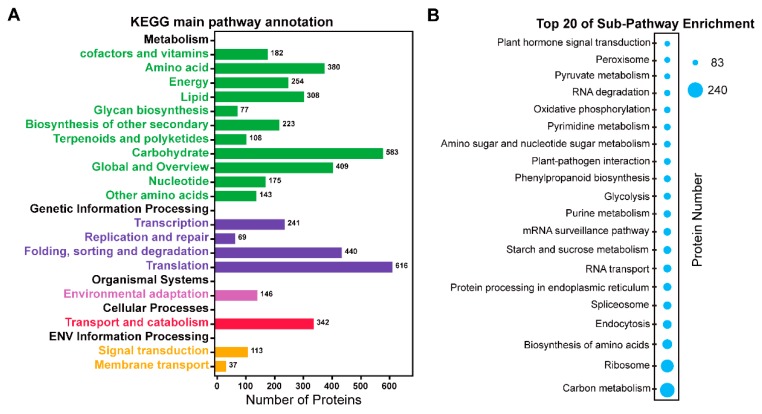
KEGG pathway analysis of all the identified proteins. KEGG pathways for the identified 3592 proteins are presented (**A**). The number of proteins enriched in the KEGG database is marked in the right side. Then, the most abundant 20 KEGG pathways are highlighted (**B**). The size of the circular ring, ranging from 83 to 240, stands for the number of proteins involved in these pathways.

**Figure 7 ijms-20-02596-f007:**
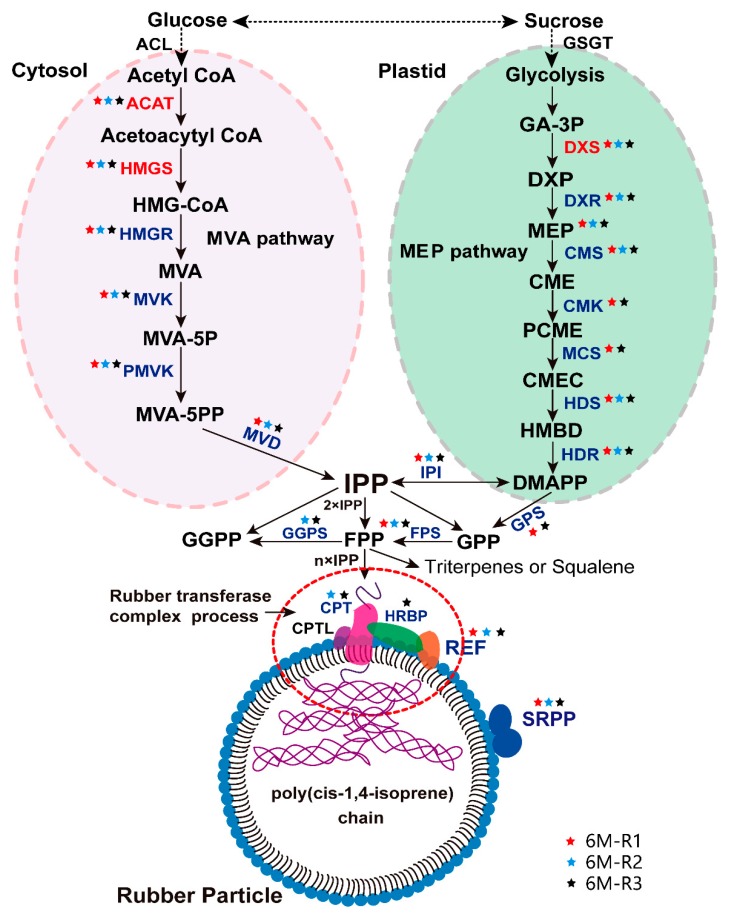
Schematic diagram of natural rubber biosynthesis (NRB)-related proteins in the 6M TKS roots. A schematic diagram of the identified proteins involved in the mevalonate acid (MVA, left) and methylerythritol phosphate (MEP, right) pathways of NRB is highlighted. In this schematic diagram, the identified proteins involved in NRB from the 6M TKS roots were marked with different color. The name of proteins identified from both 2-DE and shotgun methods was marked in red color. The proteins only determined by shotgun analysis are marked in blue color. The proteins identified from the first (6M-1), second (6M-2) and third (6M-3) shotgun experiments are marked with red, blue and black stars, respectively.

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
