# Peer review of "Proteomic Landscape of the Mature Roots in a Rubber-Producing Grass Taraxacum Kok-saghyz"

_ijms, 2019, doi:10.3390/ijms20102596_

Round 1

Reviewer 1 Report

The MS is interesting but several corrections are necessary before publication.

In fact:

-        Italics is necessary for Specie names;

-        Line 115,  Figure 2 please;

-        Line 128, Authors should consider the possibility that the procedure employed could have determined protein modifications;

-        Figure 4 A must be moved to Figure 3 as Figure 3C;

-        Figure 5: genes in the figure but proteins in the Figure legend, please correct;

-        As above for Figure 6;

-        Conclusions and Abstract: authors must clarify why the proteomic data reveal that SRPP may be more important than REF;

-        The English needs revision

Author Response

Response to Reviewer One

Comments and Suggestions for Authors

The MS is interesting but several corrections are necessary before publication.

(Response: Thank you very much for this Reviewer to give us so many detail corrections and helpful suggestions. According to the Reviewer’s helpful suggestions, we have further carefully revised this manuscript. The pointed mistakes have been corrected in the revised version. The changed parts were marked in red color in the newly revised manuscript.)

In fact:

-        Italics is necessary for Specie names;

(Response: We are very sorry for these mistakes. All specie names have been provided in Italics” format in the newly revised version.)

-        Line 115,  Figure 2 please;

(Response: We have corrected the word " Figure 1" to “Figure 2” on line 115 in the revised manuscript.)

-        Line 128, Authors should consider the possibility that the procedure employed could have determined protein modifications;

(Response: Thanks for the reviewer’s constructive comment. Actually, in the manuscript, we have analyzed that the possibility of the employed procedure that could have determined protein modifications, so we have corrected the sentence “These results indicated that these proteins could have post-modifications that resulted in protein forms/species in the 2-DE gel-based proteomics.”)

-        Figure 4 A must be moved to Figure 3 as Figure 3C;

(Response: Thanks for the reviewer’s helpful suggestion. We have moved Figure 4A to Figure 3C, and new Figures 3 and 4 have been provided. Accordingly, the figure legend and the positions for Figures 3 and 4 have been corrected in the revised manuscript.)

-        Figure 5: genes in the figure but proteins in the Figure legend, please correct;

(Response: Thank you for your kindly remind. We have replaced genes with proteins in the new Figure 5, and corrected the legend for Figure 5 in line 193 in the revised manuscript.)

-        As above for Figure 6;

(Response: Thank you for your kindly remind. We have replaced genes with proteins in the new Figure 6, and corrected the legend for new Figure 6 in line 216 in the revised manuscript.)

-        Conclusions and Abstract: authors must clarify why the proteomic data reveal that SRPP may be more important than REF;

(Response: In this manuscript for proteomics of TKS roots, eight SRPP members were identified, but only two REF members were identified in the mature roots (Table S6). In the genome of the Hevea rubber tree, eight REF members were determined, but only two REF members have been characterized in the TKS genome. These results in the genomes of TKS and Heave, as well as our proteomics of the 6M TKS roots, revealed that SRPP might be more important than REF for rubber biosynthesis in the mature roots of TKS. Therefore, based on these genomic and proteomic data, we think SRPP might be more important for natural rubber biosynthesis in TKS roots.)

-        The English needs revision

(Response: Thank you. The English language of the revised manuscript has been carefully revised by all the authors, and then polished by our friend, a native English speaker in USA, and Dr. Xuyuan Zhang, a new associate professor in our group. Some typographical errors, grammar, spelling, vocabulary, and confusing sentences were corrected. The changed parts were marked in red color in the newly revised manuscript. We think that this newly revised version might be more suitable for publication in IJMS.)

Reviewer 2 Report

The manuscript of Xie et al. represent a potent contribution to the field of natural rubber biosynthesis on TKS. The document is well structured, the most recent literature is appropriately cited, results (figures, tables and supplemental materials) are clear and concise. Material and methods section also is well documented just the brand of bovine trypsin is missing (although authors include a catalog number). I reccomend this manuscript for publication after authors revise the English style of the document, mainly because of some inappropriate word use.

Author Response

Response to Reviewer Two

Comments and Suggestions for Authors

The manuscript of Xie et al. represent a potent contribution to the field of natural rubber biosynthesis on TKS. The document is well structured, the most recent literature is appropriately cited, results (figures, tables and supplemental materials) are clear and concise.

(Response: Thank you very much for this Reviewer to review and give us so many good comments. According to the Reviewer’s helpful suggestions, we have further carefully revised this manuscript. The pointed mistakes have been corrected in the revised version.)

Material and methods section also is well documented just the brand of bovine trypsin is missing (although authors include a catalog number).

(Response: Thank you very much for pointing out this problem. In our lab for protein in-gel digestion, we always use the modified bovine trypsin (Cat. 11418025001) that produced by Roche, a global pioneer in pharmaceuticals and diagnostics. We have supplemented the brand information of the used bovine trypsin on Line 445 trypsin (Cat. 11418025001, Roche, Basel, Switzerland) in the revised manuscript.)

I recommend this manuscript for publication after authors revise the English style of the document, mainly because of some inappropriate word use.

(Response: Thank you. The English language of the revised manuscript has been carefully revised by all the authors, and then polished by our friend, a native English speaker in USA, and Dr. Xuyuan Zhang, a new associate professor in our group. Some typographical errors, grammar, spelling, vocabulary, and confusing sentences were corrected. The changed parts were marked in red color. This newly revised version might be more suitable for publication in IJMS.)

Reviewer 3 Report

The manuscript “Proteomic landscape of mature roots in the rubber producing grass Taraxacum kok-saghyz” presents a novel work with a proper research objective and methodologies. Looking at the commercial importance of the rubber grass Taraxacum kok-saghyz (TKS), this work can be considered as of immense importance. The manuscript has been prepared well and describes the methods and results extensively with a nice discussion.

Comments and suggestions for improvement:

1.     The abstract is well written with defined hypothesis and findings.

2.     Morphological analysis of TKS was done nicely with photographic evidences.

3.     The 2DE gels were explained well but there is a need of explaining more about the proteins with a difference in pI value regarding their post-modifications.

4.     In plant growth conditions, the composition of the soil/medium is not given, whether anything other than soil is used or not. If yes, then in what ratio.

5.     In conclusion mentioning of a detailed insight about the protein categories and their function would have been made the study more valuable to fellow researchers.

6.     In the Figure 5, the labels mentioning “biological process”, “cellular component” and “molecular functions” with their respective colors in the form of bars do not properly cover the related sections.

7.     The results of functional annotation such as outputs of WEGO can be mentioned in a more efficient way. It is unclear in the Results section how the outputs of this tool helped.

8.     The KEGG pathway analysis in this study was performed using the BLAST2GO program. Kindly mention in detail in the Materials and methods section.

9.     The functional annotation part under the Materials and methods section could be more descriptive.

Author Response

Response to Reviewer Three

Comments and Suggestions for Authors

The manuscript “Proteomic landscape of mature roots in the rubber producing grass Taraxacum kok-saghyz” presents a novel work with a proper research objective and methodologies. Looking at the commercial importance of the rubber grass Taraxacum kok-saghyz (TKS), this work can be considered as of immense importance. The manuscript has been prepared well and describes the methods and results extensively with a nice discussion.

(Response: Thank you very much for this Reviewer to review and give us so many good comments and helpful suggestions, we have further carefully revised this manuscript. The pointed mistakes have been corrected in the revised version. The changed parts were marked in red color in the newly revised manuscript.)

Comments and suggestions for improvement:

1. The abstract is well written with defined hypothesis and findings.

(Response: Thank you very much.)

2. Morphological analysis of TKS was done nicely with photographic evidences.

(Response: Thank you for your recognition of these morphological results.)

3. The 2DE gels were explained well but there is a need of explaining more about the proteins with a difference in pI value regarding their post-modifications.

(Response: Thank you for your valuable suggestions. In the primary proteomic study, we only determined the main proteins in the 2-DE gel and compared the distribution patterns of the experimental values of molecular weight (Mr) and isoelectric point (pI) of the 371 proteins. From these data, we noticed that many proteins demonstrated differentially experimental Mr and pI values in the 2-DE gel. In proteomics study based on 2-DE gel, they are called different protein isoforms or protein species. In this study, 90 unique proteins were identified from at least two different spots, and these proteins contained 229 spots in the 2-DE gel. These protein isoforms have the same theoretical Mr and pI values, but their experimental Mr and pI values are different with each other, and most protein isoforms showed a different pI value in the 2-DE gel (Figure 1). These results indicated that these proteins could have post-modifications that resulted in protein forms/species in the 2-DE gel-based proteomics (Table S1), probably due to the existence of modification protein variants. However, we did not perform further experiments to determine the detail post-modifications in these proteins and did not give more explains for the difference in pI value. We think more explains will be provided for these proteins only after the detail post-modifications in them have been determined in the future.)

4. In plant growth conditions, the composition of the soil/medium is not given, whether anything other than soil is used or not. If yes, then in what ratio.

(Response: Thank you for your advice. The ratio of the medium is nutritive soil: vermiculite=5:3, and 1/2 of Hoagland liquid medium was irrigated during the culture. This part has been added to the materials and methods on line 445 in the revised version.)

5. In conclusion mentioning of a detailed insight about the protein categories and their function would have been made the study more valuable to fellow researchers.

(Response: OK. Thank you for your suggestion. In the revised conclusion part, we have added more details about the protein categories and their function. The revised part is: Functional category of all TKS proteins revealed that most proteins were involved in carbon metabolic process with catalytic activity in membrane-bounded organelle, followed by proteins with binding ability, transportation activity, starch and sucrose metabolism, and phenylpropanoid biosynthesis. They are important for biosynthesis of isoprenoids in natural rubber producing plants. We completely with this Reviewer’s opinion that these new sentences may make this study more valuable to the potential readers and fellow researchers.)

6. In the Figure 5, the labels mentioning “biological process”, “cellular component” and “molecular functions” with their respective colors in the form of bars do not properly cover the related sections.)

(Response: Thank you for your suggestion. We have changed “biological process”, “cellular component” and “molecular functions” use red, blue and green color in the form of bars to distinguish the corresponding sections in the new Figure 5 of the newly revised manuscript.)

7. The results of functional annotation such as outputs of WEGO can be mentioned in a more efficient way. It is unclear in the Results section how the outputs of this tool helped.

(Response: We gratefully thank the reviewer’s suggestion. It is helpful to improve the quality of the entire article. WEGO (Web Gene Ontology Annotation Plot) is a simple but useful tool for visualizing, comparing and plotting GO (Gene Ontology) annotation results. WEGO uses the GO annotation results as input. Based on GO’s standardized DAG (Directed Acyclic Graph) structured vocabulary system, the number of genes corresponding to each GO ID is calculated and shown in a graphical format. In the revised manuscript, we mentioned the outputs of WEGO in a more efficient way, and give more explains for these GO results.)

8. The KEGG pathway analysis in this study was performed using the BLAST2GO program. Kindly mention in detail in the Materials and methods section.

(Response: Thanks for the reviewer's advice. A detailed description of Blast2GO can be easily obtained in the description of the reference No. 66. In brief, open Blast2GO software, then click file and load sequences (eg: fasta) to import data (fasta format). Select the file and type to import. Load and blast, run blast, run the database after selecting the database. After the mapping ends, the Run annotation follows. If RUN is finished, GO SLIM and run go-slim. We think the potential readers can get these skills from the NO. 66 reference, and so did not provided much more details in the Materials and methods section.)

9. The functional annotation part under the Materials and methods section could be more descriptive.

(Response: Thanks for the reviewer's suggestion. The description of the function annotations in the materials and methods has been described in detail in the UniProt database (http://www.uniprot.org/) and the Gene Ontology database (GO, http://www.geneontology.org/) as described in the reference NO. 66. We think the potential readers can get these skills from the NO. 66 reference and the two database. Therefore, we did not provided much more details in the Materials and methods section in our original manuscript.)
